# Burden and bacterial etiology of neonatal meningitis at Hawassa University Comprehensive Specialized Hospital, Hawassa, Ethiopia

**Musa Mohammed Ali** [ID] *

College of Medicine and Health Sciences, School of Medical Laboratory Science, Hawassa University, Hawassa, Ethiopia

* ysnmss@yahoo.com

## Abstract

### Background

Meningitis poses a significant challenge to public health in low-income nations, such as Ethiopia, with a particular impact on newborns. The magnitude and etiologies of meningitis vary based on geographic location and age of patients. There is limited data regarding the magnitude and etiology of meningitis from Sidama Regional State, Ethiopia. This study aimed to determine the magnitude and bacterial profile of meningitis among newborns aged less than 90 days at Hawassa University Comprehensive Specialized Hospital (HUCSH).

### Methods

A retrospective cross-sectional study was conducted among newborns under 90 days who were suspected of meningitis at HUCSH from January 2019 to July 2023, and for whom Cerebrospinal fluid (CSF) culture was performed. At HUCSH, bacteria are isolated and identified using standard microbiological techniques. Socio-demographic characteristics and culture results were extracted from the laboratory register. Data were entered into Excel and exported it to SPSS version 20 for analysis.

### Results

Overall 1061 newborns suspected of meningitis were included in the study. Among the participants, 767 individuals (72.3%) fell within the age range of 8 to 90 days. Of the total participants, 437 (41.2%) were females. The magnitude of culture-confirmed meningitis was 90 (8.5%) 95% CI: 6.8%−10.1%. The magnitude of culture-confirmed meningitis among newborns aged 0−7 days and 8−90 days were 1.6% and 6.9% respectively. The proportion of bacteria among newborns aged 0−7 days and 8−90 days were 18.9% and 81.1% respectively. Coagulase-negative Staphylococci (CONS) were the most common bacteria (n = 26; 28.9%) recovered followed by *Acinetobacter* species (n = 12, 13.3%), *Escherichia coli* (n = 9; 10%), and *Klebsiella pneumoniae* (n = 7; 7.8%). *K. pneumoniae* was the predominant bacteria among newborns within the age group of 0 to 7 days while *Acinetobacter* species

**Data Availability Statement:** All relevant data are within the paper and its Supporting Information files.

**Funding:** The author(s) received no specific funding for this work.

**Competing interests:** The authors have declared that no competing interests exist.

was the most common among newborns within the 8 to 90 days age group. The prevalence of culture-confirmed neonatal meningitis was found to be greater in male newborns ($x^2$ = 1.74, $p$ = 0.18), newborns aged between 8 to 90 days ($x^2$ = 0.07, $p$ = 3.4), and newborns admitted in 2022 ($x^2$ = 2.4, $p$ = 0.66),

## Conclusions

In this study, the overall magnitude of culture-confirmed meningitis was relatively high. Culture-confirmed meningitis was high in newborns within the age range of 8 to 90 days. The most common bacteria were CONS in both age groups followed by *Acinetobacter* species, *E. coli*, and *K. pneumoniae*.

## Introduction

Meningitis is characterized by the inflammation of the tissues surrounding the brain and spinal cord, usually caused by an infection [1]. While a small number of cases can be linked to drugs, injuries, and cancers, bacterial meningitis is the most dangerous form and can lead to death within one day [1, 2]. Approximately 20% of individuals who survive an episode of bacterial meningitis may experience persistent consequences. These consequences encompass hearing impairment, seizures, muscle weakness, visual impairments, speech and language difficulties, and memory problems [1].

Meningitis poses a significant public health challenge in Africa, particularly in Ethiopia, with a disproportionate impact on newborns. The magnitude and mortality rates differ based on geographic regions, etiologies, and patient age [2]. Even though there is a vaccine for common causes, meningitis remains a significant cause of mortality and morbidity in Africa [1–5].

There are several potential causes of meningitis, which encompass bacterial, viral, and fungal origins [6, 7]. Etiologies of meningitis can disseminate through droplets of respiratory secretions and from mother to baby during or before birth [8, 9]. *Haemophilus influenzae*, *Streptococcus pneumoniae*, and *Neisseria meningitides* are the main causative agents of meningitis in adults and children above the age of one year whereas *Streptococcus agalactiae*, *Escherichia coli*, *Listeria monocytogenes*, and *Staphylococcus aureus* are common in newborns [1]. Neonates are susceptible to contracting meningitis through various means, with the primary routes being vertical transmission, healthcare-related transmission, and community-acquired transmission. For instance, neonates may contract bacterial meningitis, such as Group B Streptococcus (GBS) or *E. coli*, from their mothers during pregnancy, labor, or delivery as a result of maternal colonization with these pathogens [10].

Two studies from Ethiopia reported a culture-confirmed magnitude of meningitis less than 5% [11, 12]. The magnitude of neonatal meningitis at Tikur Anbessa Specialized Hospital in Addis Ababa, Ethiopia was recorded to be 4.7%, with predominant causes being *S. pneumoniae* and *E. coli*. [11]. The magnitude of neonatal meningitis at Gondar, Ethiopia was 1.73%, and the predominant cause was *Klebsiella pneumoniae* [12].

Meningitis can be categorized into two types: early-onset and late-onset. Early-onset meningitis is seen in newborns who are 7 days old or less, while late-onset meningitis is observed in newborns aged 8 to 90 days [13–17]. The vulnerability of newborns to meningitis is attributed to their underdeveloped humoral and cellular immunity, making it difficult for them to combat infections. Additionally, if the mother carries the pathogen during pregnancy, newborns can acquire the infection before birth.

Meningitis can be life-threatening or result in a neurologic defect [16, 17]. Death and neuropsychological sequelae from meningitis are higher in low-income countries because of a lack of diagnostic setup and facilities to manage the cases as early as possible [18, 19]. Non-specific signs and symptoms of meningitis and cardiorespiratory instability in neonates may delay the diagnosis and treatment, which results in an increased incidence of mortality [14, 20, 21].

Even though meningitis is among high-priority public health problems, there is limited evidence in Ethiopia in particular there is no study from Sidama Regional State reporting the magnitude and etiologies of neonatal meningitis. This study aimed to determine the magnitude and etiology of neonatal meningitis among newborns who sought treatment at Hawassa University Comprehensive Specialized Hospital (HUCSH).

## Methods

### Study area and design

A retrospective cross-sectional study was conducted at HUCSH, a hospital located in Hawassa City, the capital of the newly formed Sidama Regional State in Ethiopia. Data collection period was from14/10/2023 to 14/11/2023. Data was collected from patients who were admitted to HUCSH within time interval of 2019 to 2023. Hawassa City is situated approximately 273 km south of Addis Ababa, capital City of Ethiopia. HUCSH was established in November 2005 and provides services to a population of around 12 million people. The hospital offers medical services to patients through various outpatient and inpatient units, including surgery, gynecology and obstetrics, internal medicine, pediatrics, ophthalmology, psychiatry, radiology, and pathology. HUCSH boasts a team of 168 clinicians, 496 nurses, 67 laboratory professionals, 67 pharmacy professionals, 78 midwives, and 52 other staff members. As a referral center, the hospital serves both public and private hospitals in the Sidama region, as well as neighboring regions such as the South Nations Nationalities People Regional State and the Oromia region.

### Study population

In the present study, all newborns under 90 days of age who were suspected of having meningitis and underwent culture testing at the microbiology laboratory of HUCSH during the period from 2019 to 2023 were included.

### Study participants and data collection

For this study, a structured format that contains age, sex, year of diagnosis, and culture results was used to extract data related to participants from the laboratory register. The etiology of meningitis is identified at HUCSH using conventional microbiological methods. In brief, Cerebrospinal fluid (CSF) specimens from newborns suspected of meningitis are collected in sterile containers by attending physicians and delivered to the microbiology laboratory within one hour. CSF is centrifuged and the sediment is inoculated onto Blood agar, Chocolate agar, and MacConkey agar to isolate bacteria. The inoculated culture plates are incubated at 35–37°C overnight in a 5% to 10% carbon-dioxide enriched atmosphere. Bacteria are identified using standard bacteriological techniques. Initially bacteria isolates were checked for colony morphology, Gram reaction, and biochemical tests. Catalase, Christie, Atkins, and Munch-Peterson (CAMP), Bacitracin, Bile solubility, and Optochin susceptibility tests were used to identify Gram-positive bacteria. To identify Gram-negative bacteria, carbohydrate fermentation, indole production, urease, motility, and Methyl Red, and Voges-Proskauer tests were used.

## Data quality and analysis

Data was collected in a pre-prepared structured format to keep the consistency of data. The collected data was verified by cross-checking 10% of randomly selected participants. Data was analyzed using SPSS version 20 software. Quantitative data was processed using cross-tabulation & frequency counts and the result of the study is presented using text and tables.

## Ethical consideration

The study was approved by the Institutional Review Board of Hawassa University College of Medicine and Health Sciences (reference number: IRB/395/15). Since the study is retrospective informed consent from guardians/parents was not sought.

## Results

Overall 1061 newborns suspected of meningitis were included in this study. The age of 294 (27.7%) participants were less than 8 days, while 767 (72.3%) were within the age range of 8 to 90 days. Of the total participants, 437 (41.2%) were females and 624 (58.8%) were males. The magnitude of culture-confirmed meningitis was 90(8.5%) 95% CI: 6.8%−10.1%. The magnitude of culture-confirmed meningitis excluding CONS was 6%.

The magnitude of culture-confirmed meningitis among newborns aged 0−7 days and 8−90 days was 17/1061(1.6%) and 73/1061(6.9%) respectively. The proportion of bacteria among newborns aged 0−7 days and 8−90 days were 17/90 (18.9%) and 73/90 (81.1%) respectively. The magnitude of culture-confirmed meningitis was high among females and among those who were within the age range of 8 to 90 days. The highest prevalence of culture-confirmed meningitis was observed in the year 2023 (n = 28; 9.8%) whereas the least prevalence was observed in the year 2019 (n = 12; 6%). The prevalence of culture-confirmed neonatal meningitis was found to be greater in male newborns ($x^2$ = 1.74, $p$ = 0.18), newborns aged between 8 to 90 days ($x^2$ = 0.07, $p$ = 3.4), and newborns admitted in 2022 ($x^2$ = 2.4, $p$ = 0.66), although these associations were not deemed statistically significant. did significantly vary across sex, year of admission, and age group ($p$>0.05) (Table 1). The most common bacteria recovered was CONS (n = 26; 28.9%) followed by *Acinetobacter* species (n = 12; 13.3%), *E. coli* (n = 9; 10%), and *K. pneumoniae* (n = 7; 7.8%). *Klebsiella pneumoniae* was the most common bacteria among newborns within the age group of 0 to 7 days while *Acinetobacter* species was the most common among newborns within the 8 to 90 days age group (Table 2).

**Table 1. Characteristics of newborns suspected of meningitis at Hawssa University Comprehensive Specialized Hospital, 2024.**

| Variables | Category | Total participants | Total culture-confirmed result (n = 90) | Culture result n (%) | | Chi-square | P-value |
|---|---|---|---|---|---|---|---|
| | | Frequency (%) | | Positive | Negative | | |
| Sex | Male | 624 (58.8%) | 47 (52.2) | 47 (7.5) | 577 (92.5) | 1.74 | 0.18 |
| | Female | 437 (41.2%) | 42 (46.7) | 43 (9.8) | 394 (90.2) | | |
| Age group | 0−7 days | 294 (27.7) | 17 (18.9) | 17 (5.8) | 277 (94.2) | 3.4 | 0.07 |
| | 8−90 days | 767 (72.3) | 73 (81.1) | 73 (9.5) | 694 (90.5) | | |
| Year visited the hospital | 2019 | 200 (18.9) | 11 (12.2) | 12 (6) | 189 (94) | 2.4 | 0.66 |
| | 2020 | 264 (24.9) | 23 (25.6) | 23 (8.7) | 241 (91.3) | | |
| | 2021 | 197 (18.6) | 17 (18.9) | 17 (8.6) | 180 (91.4) | | |
| | 2022 | 285 (26.9) | 28 (31.1) | 28 (9.8) | 257 (90.2) | | |
| | 2023 | 115 (10.8) | 10 (11.1) | 10 (8.7) | 105 (91.3) | | |

**Table 2. Distribution of bacteria across age groups among newborns suspected of meningitis at Hawssa University Comprehensive Specialized Hospital, 2024.**

| Type of bacteria | | Frequent bacteria n (%) | | Proportion of bacteria in age groups n (%) | |
|---|---|---|---|---|---|
| | Total n (%) | Age 0–7 days | Age 8–90 days | 0–7 days | 8–90 days |
| *Acinetobacter* species | 12 (13.3) | 1 (5.9) | 11 (15) | 1 (8.3) | 11 (91.7) |
| CONS | 26 (27.8) | 10 (58.8) | 16 (21.9) | 10 (38.5) | 16 (61.5) |
| *Enterococcus* species | 7 (7.8) | – | 7 (9.6) | – | 7 (100) |
| *E. coli* | 9 (10) | 1 (5.9) | 8 (10.9) | 1 (11.1) | 8 (88.9) |
| GPR | 2 (2.2) | – | 2 (2.7) | – | 2 (100) |
| *Klebsiella oxytoca* | 3 (2.2) | – | 3 (4.1) | – | 3 (100) |
| *Klebsiella ozaenae* | 1 (1.1) | – | 1 (1.4) | – | 1 (100) |
| *Klebsiella pneumoniae* | 7 (7.8) | 2 (11.8) | 5 (6.9) | 2 (28.6) | 5 (71.4) |
| *Neisseria meningitis* | 1 (1.1) | – | 1 (1.4) | – | 1 (100) |
| NLF GNR | 2 | 1 (5.9) | 1 (1.4) | 1 (50) | 1 (50) |
| *Pseudomonas* species | 7 (7.8) | 1 (5.9) | 6 (8.2) | 1 (14.3) | 6 (85.7) |
| *S. aureus* | 5 (4.4) | – | 5 (6.9) | – | 5 (100) |
| *S. lugdunensis* | 5 (4.4) | – | 5 (6.9) | – | 5 (100) |
| *Streptococcus viridians* | 1 (1.1) | – | 1 (1.4) | – | 1 (100) |
| Yeast cells | 2 (1.1) | 1 (5.9) | 1 (1.4) | 1 (50) | 1 (50) |
| **Total** | **90** | **17** | **73** | **17** | **73** |

GPR: Gram positive rod, NLF: None lactose fermenter, GNR: Gram negative rod, CONS: Coagulase Negative Staphylococcus

## Discussion

Meningitis is a matter of great concern due to its severe consequences. It has a high mortality rate, with approximately 1 in 6 individuals succumbing to the disease, and 1 in 5 experiencing severe complications. The magnitude and fatality rates of bacterial meningitis differ across regions, countries, pathogens, and age groups. Low-income countries face a considerable threat from bacterial meningitis, as it is linked to a high rate of death and the development of long-term complications. Africa bears the highest magnitude of bacterial meningitis, with outbreaks occurring seasonally and varying based on geographic location [17, 20].

In this study, the magnitude of culture-confirmed meningitis was 90(8.5%) 95% CI: 6.8% −10.1%. The results of this study indicate a higher percentage compared to a study conducted in Addis Ababa, Ethiopia, which reported a rate of 4.7% [11], and Gondar, which reported a rate of 1.73% [12]. Several factors could account for the observed discrepancy, such as variations in the laboratory techniques employed, seasonal variations, the presence of underlying conditions among the study participants, and the specific types of bacteria that were taken into consideration. It is worth noting that in this particular study, coagulase-negative staphylococci (CONS) were included in the analysis to determine the overall magnitude. CONS are typically considered as contaminants, but due to the retrospective nature of the study, we cannot definitively confirm this. However, even if CONS were to be excluded from the analysis, the prevalence of culture-confirmed meningitis would still be 6%, which remains higher than the findings of previous studies conducted in Addis Ababa [11] and Gondar [12].

The highest prevalence of culture-confirmed meningitis was observed among female participants, newborns aged 8 to 90 days, and individuals admitted in the year 2023, although this variation did not reach statistical significance ($p > 0.05$). It is important to highlight that the lower number of meningitis cases in 2019 may be linked to the lockdown measures enforced during the COVID-19 pandemic. This discovery is consistent with a previous study carried out in Addis Ababa [11] and Gondar [12], which revealed that 89.5% of positive cerebrospinal fluid (CSF) cultures were identified in late-onset neonates.

This study indicated that Gram-negative and Gram-positive bacteria were responsible for meningitis in the study area. CONS were the most common bacteria (n = 26; 27.8%) recovered from newborns suspected of meningitis. Some bacteria were not identified at the species level posing challenges in analyzing the data because of lack of supplies in the study area. The most prevalent bacteria next to CONS among both age groups was *Acinetobacter* species (13.3%) followed by *E. coli* (10%) and *K. pneumoniae* (7.8%). *K. pneumoniae* was the most common bacteria among newborns within the age group of 0 to 7 days while *Acinetobacter* species was the most common among newborns within the 8 to 90 days age group emphasizing both vertical and horizontal transmission of bacteria.

Most bacteria are acquired after 7 days of life indicating that they are horizontally acquired from healthcare settings or family members. However, a significant proportion of CONS were acquired before 7 days of life indicating they could be acquired from the mother during or before birth. Bacteria such as *Enterococcus* species, GPR, *Klebsiella oxytoca*, *Klebsiella ozaenae*, *Neisseria meningitis*, *S. aureus*, *S. lugdunensis*, and *Streptococcus viridians* were not isolated from newborns younger than a week. Identifying the route of transmission of bacteria (vertical or horizontal) may assist in developing appropriate prevention methods.

Similar to the current study, a study from Gondar detected Gram-negative and Gram-positive bacteria from newborns with meningitis reporting *K. pneumoniae* and Non lactose fermenter Gram-negative rods as the leading cause of meningitis. Unlike this study, GBS were the leading bacterial isolates from the Gondar [12]. Studies carried out in Iran [22, 23] and Mexico [24] have shown that *K. pneumoniae* is the primary causative agent of meningitis in newborns.

In contrast to our study, a study conducted in Addis Ababa, Ethiopia, revealed that *S. pneumoniae* was identified as the primary cause of meningitis, followed by *E. coli* and *Acinetobacter*. Additionally, *N. meningitides*, *Klebsiella* species, *S. aureus*, *S. pyogenes*, and CONS were also reported as role players of meningitis [11]. A systematic review and meta-analysis conducted in sub-Saharan Africa revealed that GBS is the primary cause of meningitis in the region [25]; however, this is not repeated in the present study. On the other hand, the magnitude of *S. pneumoniae* as the most prevalent bacteria was highlighted in a study carried out by the World Health Organization (WHO) [26]. As this study is retrospective it is difficult to verify the data and assess the quality of the laboratory method used, and also additional information was not captured in the record to do association.

## Conclusions

The overall magnitude of culture-confirmed meningitis was high. Culture-confirmed meningitis was high among newborns within the age range of 8 to 90 days. The highest prevalence of culture-confirmed meningitis was observed in the year 2023, in females, and newborns aged 8 to 90 days. The most common bacteria were CONS followed by *Acinetobacter* species, *E. coli*, and *K. pneumoniae*.

## Supporting information

**S1 Raw data.**
(SAV)

## Author Contributions

**Conceptualization:** Musa Mohammed Ali.

**Data curation:** Musa Mohammed Ali.

**Formal analysis:** Musa Mohammed Ali.

**Investigation:** Musa Mohammed Ali.

**Methodology:** Musa Mohammed Ali.

**Project administration:** Musa Mohammed Ali.

**Resources:** Musa Mohammed Ali.

**Supervision:** Musa Mohammed Ali.

**Validation:** Musa Mohammed Ali.

**Visualization:** Musa Mohammed Ali.

**Writing – original draft:** Musa Mohammed Ali.

**Writing – review & editing:** Musa Mohammed Ali.

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
