## [Decision Letter · Decision Letter 0]

5 Jun 2024

PONE-D-24-15982Burden and etiology of neonatal meningitis at Hawassa University Comprehensive Specialized HospitalPLOS ONE

Dear Dr. Ali,

Thank you for submitting your manuscript to PLOS ONE. After careful consideration, we feel that it has merit but does not fully meet PLOS ONE’s publication criteria as it currently stands. Therefore, we invite you to submit a revised version of the manuscript that addresses the points raised during the review process.

We look forward to receiving your revised manuscript.

Kind regards,

Tebelay Dilnessa, MSc

Academic Editor

PLOS ONE

2. We note that your Data Availability Statement is currently as follows: [Author declares that there is no competing interest.]

Additional Editor Comments:

**Dear Author,**

I acknowledged author's work entitled: Burden and etiology of neonatal meningitis at Hawassa University Comprehensive Specialized Hospital. Even though, a single author the results illustrated the burden of meningitis among children*.* Therefore, this study will be useful for prevention and control of bacterial meningitis among children in the study area. The manuscript is more or less in line with the vision, objectives and instructions of PLoS One.

**Title: **‘Burden and etiology of neonatal meningitis at Hawassa University Comprehensive Specialized Hospital’ It is better writer as, ‘Burden and bacterial etiology of neonatal meningitis at Hawassa University Comprehensive Specialized Hospital, Hawassa, Ethiopia’

**Abstract**

Lines 9: ‘Introduction’ better replaced by ‘Background’.

**Introduction**

Line 82: …………………HUCSH, better written as, Hawassa University Comprehensive Specialized Hospital (HUCSH).

**Materials and Methods **

No sample size determination and sampling technique was described.Line 101: ‘Bacteria were identified using standard bacteriological techniques. Which bacteriological technique?Specimen collection and processing needs to be well written.What standard microbiological procedures were employed in specimen collection, isolation, identification susceptibility testing and throughout the procedure?How could you standardize the suspension preparation during performing antimicrobial susceptibility testing?An ethical approval was also needed. What have you done for children positive for the microorganisms?

**Results**

Table descriptions should be described well in terms of person, place and time.Line 124: Table 2, words/phrases like: ‘CONS, GPR, NLF GNR’ requires legend or a full form as they appear for the first time.Please follow the manuscript writing protocol of PLOS ONE for figure preparation and presentation.What is your base to classify the age group?Why you missed to include the associated risk factors?
Authors tried to compare their findings with different reports from worldwide. They also better justify the reason for variation among results of different research findings with respect to theirs’ based on actual situation.

**Conclusions**

Normally, conclusion emanates from the research.

Reviewers' comments:

Reviewer's Responses to Questions

**Comments to the Author**

1. Is the manuscript technically sound, and do the data support the conclusions?

Reviewer #1: Yes

Reviewer #2: Yes

2. Has the statistical analysis been performed appropriately and rigorously? 

Reviewer #1: Yes

Reviewer #2: No

3. Have the authors made all data underlying the findings in their manuscript fully available?

Reviewer #1: Yes

Reviewer #2: Yes

4. Is the manuscript presented in an intelligible fashion and written in standard English?

Reviewer #1: Yes

Reviewer #2: Yes

5. Review Comments to the Author

Reviewer #1: The title "Burden and etiology of neonatal meningitis at Hawassa University Comprehensive

Specialized Hospital" needs modification" because fungal and viral agents are not considered throughout the manuscript. Simply modify the title, like the aim explained in the abstract section (L13–15).  Lines 13–15 and 16–18,  in the introduction and methods sections of the abstract, respectively, reveal different participant inclusions. A similar source population must be used, whereas all newborns less than 90 days of age and suspected newborns are explained, respectively. Moreover, it is stated that "At HUCSH, isolation and identification of

19 bacteria were performed using standard microbiological techniques". Since it is retrospective, it must be explained in a different way. In the results of the abstract section, "Line 1: The age of 767 (72.3%) was between 8 and 90 days" is unclear. L52-53 "Among the various etiologies of meningitis, bacteria are responsible for the highest burden.

53 worldwide" is unrealistic. What about viruses? L69-71: "Death and neuropsychological sequelae from meningitis are higher in low-income countries because of a lack of facilities to manage the cases as early as possible." What about other very important factors? Additionally, words like burden, magnitude, and prevalence are not used consistently at the beginning of your research question, and it is better to use only one of them throughout the document. Consider L17 VS 83. Are you studying infants (L-89) or newborns? Line 91: silly mistake, the year (needs correction). Lots of information is missed in data collection, transportation, handling, and data quality control. L130 to L132 "This life-threatening illness poses a significant risk, particularly in low-income countries, where it is associated with a high case fatality rate and long-term complications," needs rewriting (similar meaning by different phrases).

Generally, there are some typos and grammar issues to accept after minor revision.

Reviewer #2: General comments

The topic is very interesting and concerned with vulnerable study group. However the manuscript prepared as simple description. for example the author considered sex, age and the year of visit as independent variable but not tried to see the relationship between the outcome variable and independent variables. therefore much effort is needed to improve the manuscript.

specific comments

Abstract

Result:

#27-28 Coagulase- negative Staphylococci (CONS) were the most common agents of meningitis. (Please include the percentage and absolute no)

#29 Acinetobacter species (n=12.2; 13.3%) followed by Escherichia coli (n=9; 10%) and Klebsiella pneumonia. ‘n’ stands for absolute number, so it should be corrected

Conclusion:

#34-35 The overall magnitude of culture-confirmed meningitis was high. What is your based to say high? Have you compared with previous study in Ethiopia or any other country considering the confidence interval? Otherwise difficult to say high or low.

Introduction

No overall data related to the burden were documented at national level. Please consider to include it. You may get it either from EDHS or from systematic review studies.

Methods

Study design and area

Please mention the catchment area of the Hospital and the environmental factors that might be associated with meningitis. Please also write the detail of Hospital service and the number of clients seeking services.

#91 from 2019 to 202. Kindly correct this figure

Study participants and data collection

Please briefly explain how the sumple collected and how the isolation and identification carried out.

Data analysis and interpretation

Your study is simple descriptive. On the other hand your finding compared with other findings. Though you reported demographic characteristic, you didn’t associated your finding. Therefore I kindly revise your result accordingly.

It would be better if you show the trend of meningitis burden across the year too.

Discussion

The discussion part needs detail explanation, Justification needs scientific ground.

6. PLOS authors have the option to publish the peer review history of their article (what does this mean?). If published, this will include your full peer review and any attached files.

Reviewer #1: **Yes: **Zigale Hibstu

Reviewer #2: No

---

## [Author Response · Author response to Decision Letter 0]

25 Jun 2024

PONE-D-24-15982

Burden and etiology of neonatal meningitis at Hawassa University Comprehensive Specialized Hospital

PLOS ONE

Dear Dr. Ali,

Thank you for submitting your manuscript to PLOS ONE. After careful consideration, we feel that it has merit but does not fully meet PLOS ONE’s publication criteria as it currently stands. Therefore, we invite you to submit a revised version of the manuscript that addresses the points raised during the review process.

We look forward to receiving your revised manuscript.

Kind regards,

Tebelay Dilnessa, MSc

Academic Editor

PLOS ONE

Response: I have checked and corrected accordingly. 

2. We note that your Data Availability Statement is currently as follows: [Author declares that there is no competing interest.]

Response: I have uploaded dataset as supplement file. 

Response: I have removed figure 1 from the manuscript. 

Additional Editor Comments:

Dear Author,

I acknowledged author's work entitled: Burden and etiology of neonatal meningitis at Hawassa University Comprehensive Specialized Hospital. Even though, a single author the results illustrated the burden of meningitis among children. Therefore, this study will be useful for prevention and control of bacterial meningitis among children in the study area. The manuscript is more or less in line with the vision, objectives and instructions of PLoS One.

Title: ‘Burden and etiology of neonatal meningitis at Hawassa University Comprehensive Specialized Hospital’ It is better writer as, ‘Burden and bacterial etiology of neonatal meningitis at Hawassa University Comprehensive Specialized Hospital, Hawassa, Ethiopia’

Abstract

• Lines 9: ‘Introduction’ better replaced by ‘Background’.

Response: We greatly appreciate your valuable feedback and have made the necessary revisions accordingly. The manuscript now reflects all the corrections, which are clearly highlighted using the track change feature.

Introduction

• Line 82: …………………HUCSH, better written as, Hawassa University Comprehensive Specialized Hospital (HUCSH).

Response: Thank you for the comment. I have followed the suggested instructions and written the text accordingly.

Materials and Methods

• No sample size determination and sampling technique was described.

Response: We appreciate your feedback. The research conducted was a retrospective study covering a period of five years, from 2019 to 2023. The study encompassed all newborns who were admitted to the hospital during this timeframe and met the specified inclusion criteria. Since all newborns who visited HUCSH between 2019 and 2023 were considered in the study, we determined that a sample size calculation was unnecessary.

• Line 101: ‘Bacteria were identified using standard bacteriological techniques. Which bacteriological technique?

Response: We appreciate your comment. As this study is retrospective in nature, we acknowledge that we have no control over the laboratory method employed. This limitation is duly acknowledged in the discussion section of the study. Nevertheless, we have provided a concise overview of the sample collection, processing, and identification methods utilized within the hospital setting.

• Specimen collection and processing needs to be well written.

Response: Thank you for the comment. We have provided a concise description of the specimen collection method employed in the hospital. However, it is important to note that since this study is retrospective in nature, our involvement did not extend to specimen collection, processing, and bacterial identification.

• What standard microbiological procedures were employed in specimen collection, isolation, identification susceptibility testing and throughout the procedure?

Response: Thank you for the comment. In the case of suspected meningitis at HUCSH, a sample of 1 to 5 ml of cerebrospinal fluid (CSF) is collected from the patients and sent to the microbiology lab for further examination. Once in the lab, the CSF in the first tube is centrifuged and the resulting sediment is then inoculated onto culture media for analysis. However, it is important to note that in this particular study, these procedures were not carried out in their entirety due to its retrospective nature. Instead, a summary of the procedures that were performed has been included.

How could you standardize the suspension preparation during performing antimicrobial susceptibility testing?

Response: Your insight is greatly appreciated. Nevertheless, the manuscript does not present AST data as it may not be obligatory to describe the AST in the method section.

An ethical approval was also needed. What have you done for children positive for the microorganisms?

Response: We have uploaded ethical clearance during initial submission. 

Results

o Table descriptions should be described well in terms of person, place and time.

Response: We have modified accordingly. 

o Line 124: Table 2, words/phrases like: ‘CONS, GPR, NLF GNR’ requires legend or a full form as they appear for the first time.

Response: We have corrected accordingly 

o Please follow the manuscript writing protocol of PLOS ONE for figure preparation and presentation.

Response: Thank you for suggestion, we have corrected the tables based on your suggestion. 

o What is your base to classify the age group?

Response: Thank you for the comment. Bacteria transmission methods have been categorized by us. Meningitis that manifest in newborns under 7 days old are contracted vertically from the mother, while those occurring in infants older than 7 days are acquired horizontally from the hospital, family members, and healthcare workers.

o Why you missed to include the associated risk factors?

Response: Thank you for the comment. Due to the retrospective nature of this study, we were unable to obtain the necessary variables from the record. Our analysis focused on age group, sex, and year of admission to determine if there was any association with culture-confirmed meningitis. However, none of these variables showed a significant association. To provide further information, we have included chi-square values with their corresponding p-values in table 1.

• Authors tried to compare their findings with different reports from worldwide. They also better justify the reason for variation among results of different research findings with respect to theirs’ based on actual situation.

Response: Thank you for the comment. We have revised accordingly. 

Conclusions

• Normally, conclusion emanates from the research.

Response: Our research revealed that the prevalence of culture-confirmed meningitis was significantly higher when compared to various studies carried out in Ethiopia and abroad. This observation was clearly reflected in our final analysis.

Reviewers' comments:

Reviewer's Responses to Questions

Comments to the Author

1. Is the manuscript technically sound, and do the data support the conclusions?

Reviewer #1: Yes

Reviewer #2: Yes

2. Has the statistical analysis been performed appropriately and rigorously?

Reviewer #1: Yes

Reviewer #2: No

3. Have the authors made all data underlying the findings in their manuscript fully available?

Reviewer #1: Yes

Reviewer #2: Yes

4. Is the manuscript presented in an intelligible fashion and written in standard English?

PLOS ONE does not copyedit accepted manuscripts, so the language in submitted articles must be clear, correct, 

---

## [Decision Letter · Decision Letter 1]

16 Jul 2024

PONE-D-24-15982R1Magnitude and etiologies of neonatal meningitis at Hawassa University Comprehensive Specialized HospitalPLOS ONE

Dear Dr. Ali,

Thank you for submitting your manuscript to PLOS ONE. After careful consideration, we feel that it has merit but does not fully meet PLOS ONE’s publication criteria as it currently stands. Therefore, we invite you to submit a revised version of the manuscript that addresses the points raised during the review process.

We look forward to receiving your revised manuscript.

Kind regards,

Tebelay Dilnessa, MSc

Academic Editor

PLOS ONE

Journal Requirements:

Additional Editor Comments:The author significantly improves the paper, still it requires a line by line through proofreading and make a correction in terms of punctuation and editorial issue.Title: Burden and etiology of neonatal meningitis at Hawassa University Comprehensive Specialized Hospital’ It is better written as, ‘Burden and bacterial etiology of neonatal meningitis at Hawassa University Comprehensive Specialized Hospital, Hawassa, Ethiopia.’ If you have any reason not to revise title you can raise the reason.Add some analysis techniques (Chi-square, P-value, etc) in the abstract as well as in the methods part.Line 82: Study area designLine 220: Data availability statement

Reviewers' comments:

Reviewer's Responses to Questions

**Comments to the Author**

1. If the authors have adequately addressed your comments raised in a previous round of review and you feel that this manuscript is now acceptable for publication, you may indicate that here to bypass the “Comments to the Author” section, enter your conflict of interest statement in the “Confidential to Editor” section, and submit your "Accept" recommendation.

Reviewer #1: All comments have been addressed

Reviewer #2: All comments have been addressed

2. Is the manuscript technically sound, and do the data support the conclusions?

Reviewer #1: Yes

Reviewer #2: Yes

3. Has the statistical analysis been performed appropriately and rigorously? 

Reviewer #1: Yes

Reviewer #2: Yes

4. Have the authors made all data underlying the findings in their manuscript fully available?

Reviewer #1: Yes

Reviewer #2: Yes

5. Is the manuscript presented in an intelligible fashion and written in standard English?

Reviewer #1: Yes

Reviewer #2: Yes

6. Review Comments to the Author

Reviewer #1: The authors have adequately addressed comments raised in a previous round of review the and I feel that this manuscript is now acceptable for publication. The manuscript technically soundful and the data support the conclusions. The manuscript is presented in an intelligible fashion and written in standard English.

Reviewer #2: Title

Line 1: Magnitude and Bacterial profile…….

Abstract

Line 13: Magnitude and Bacterial profile…….

Introduction

Please make one paragraph on how the new borne acquire the infection

Line 40-41: The sentence need citation

Line 41-42: The sentence need citation

Methods

Line 82: Study area desin – correct spelling error

Line 85: The study time or period should be the first date of the date record up to the last date of data record.

Line 86: south of Addis Ababa, capital City of Ethiopia

Results

Line 138: age group of 0 to 8 days, correct the range

Line 132-133: The magnitude of culture-confirmed meningitis was high among females and among those who were within the age range of 8 to 90 days. However your analysis indicated the p value is greater than 0.05 (0.18). is it possible to higher or lower? Please recheck it?

7. PLOS authors have the option to publish the peer review history of their article (what does this mean?). If published, this will include your full peer review and any attached files.

Reviewer #1: **Yes: **Zigale Hibstu Teffera

Reviewer #2: No

---

## [Author Response · Author response to Decision Letter 1]

19 Jul 2024

Response to reviewers and editor comments

Additional Editor Comments:

• The author significantly improves the paper, still it requires a line by line through proofreading and make a correction in terms of punctuation and editorial issue.

Response: Thank you again, I have re-checked the manuscript and corrected issues related to punctuations and editorials as shown in the manuscript with track change. 

• Title: Burden and etiology of neonatal meningitis at Hawassa University Comprehensive Specialized Hospital’ It is better written as, ‘Burden and bacterial etiology of neonatal meningitis at Hawassa University Comprehensive Specialized Hospital, Hawassa, Ethiopia.’ If you have any reason not to revise title you can raise the reason.

Response: Thank you for the comment; I have replaced the title with the suggested one. The change is shown in the manuscript with track change. 

• Add some analysis techniques (Chi-square, P-value, etc) in the abstract as well as in the methods part.

Response: I have included chi-square and p-value as suggested in the abstract and method. 

• Line 82: Study area design

Response: Thank you again for pointing the error, I have corrected it. 

• Line 220: Data availability statement

Response: I have corrected as suggested. 

Reviewers' comments:

Reviewer's Responses to Questions

Comments to the Author

1. If the authors have adequately addressed your comments raised in a previous round of review and you feel that this manuscript is now acceptable for publication, you may indicate that here to bypass the “Comments to the Author” section, enter your conflict of interest statement in the “Confidential to Editor” section, and submit your "Accept" recommendation.

Reviewer #1: All comments have been addressed

Reviewer #2: All comments have been addressed

2. Is the manuscript technically sound, and do the data support the conclusions?

Reviewer #1: Yes

Reviewer #2: Yes

3. Has the statistical analysis been performed appropriately and rigorously?

Reviewer #1: Yes

Reviewer #2: Yes

4. Have the authors made all data underlying the findings in their manuscript fully available?

Reviewer #1: Yes

Reviewer #2: Yes

5. Is the manuscript presented in an intelligible fashion and written in standard English?

Reviewer #1: Yes

Reviewer #2: Yes

6. Review Comments to the Author

Reviewer #1: The authors have adequately addressed comments raised in a previous round of review the and I feel that this manuscript is now acceptable for publication. The manuscript technically soundful and the data support the conclusions. The manuscript is presented in an intelligible fashion and written in standard English.

Response: Thank you for checking and approving the change made. 

Reviewer #2: Title

Line 1: Magnitude and Bacterial profile…….

Response: Thank you for the comment; I have corrected the title based on the editor comment. 

Abstract

Line 13: Magnitude and Bacterial profile…….

Response: Thank you for the comment I have corrected accordingly

Introduction

Please make one paragraph on how the new borne acquire the infection

Response: I have included a paragraph on how the newborns acquire the infection. 

Line 40-41: The sentence need citation

Response: I have included the citation as suggested. 

Line 41-42: The sentence need citation

Response: I have included the citation as suggested.

Methods

Line 82: Study area desin – correct spelling error

Response: I have corrected it. 

Line 85: The study time or period should be the first date of the date record up to the last date of data record.

Response: Thank you for the comment, I believe both times (time of data collection and admission to the hospital are important). I modified to indicate both time intervals as “Data collection period was from14/10/2023 to 14/11/2023. Data was collected from patients who were admitted to HUCSH within time interval of 2019 to 2023.” 

Line 86: south of Addis Ababa, capital City of Ethiopia

Response: I have corrected accordingly. 

Results

Line 138: age group of 0 to 8 days, correct the range

Response: I have corrected it.

Line 132-133: The magnitude of culture-confirmed meningitis was high among females and among those who were within the age range of 8 to 90 days. However your analysis indicated the p value is greater than 0.05 (0.18). is it possible to higher or lower? Please recheck it?

Response: Thank you for the comment. The proportion is higher among females (9.8% vs 7.5%) and among those who were within the age range of 8 to 90 days (9.5% vs 5.8%) but these differences are not statistically significant as p value in both cases is greater than the cut point which 0.05 

7. PLOS authors have the option to publish the peer review history of their article (what does this mean?). If published, this will include your full peer review and any attached files.

Do you want your identity to be public for this peer review? For information about this choice, including consent withdrawal, please see our Privacy Policy.

Reviewer #1: Yes: Zigale Hibstu Teffera

Reviewer #2: No

---

## [Decision Letter · Decision Letter 2]

25 Jul 2024

Burden and bacterial etiology of neonatal meningitis at Hawassa University Comprehensive Specialized Hospital, Hawassa, Ethiopia

PONE-D-24-15982R2

Dear Dr. Ali,

We’re pleased to inform you that your manuscript has been judged scientifically suitable for publication and will be formally accepted for publication once it meets all outstanding technical requirements.

Kind regards,

Tebelay Dilnessa, MSc

Academic Editor

PLOS ONE

Additional Editor Comments (optional):

Reviewers' comments:

Reviewer's Responses to Questions

**Comments to the Author**

1. If the authors have adequately addressed your comments raised in a previous round of review and you feel that this manuscript is now acceptable for publication, you may indicate that here to bypass the “Comments to the Author” section, enter your conflict of interest statement in the “Confidential to Editor” section, and submit your "Accept" recommendation.

Reviewer #2: All comments have been addressed

2. Is the manuscript technically sound, and do the data support the conclusions?

Reviewer #2: Yes

3. Has the statistical analysis been performed appropriately and rigorously? 

Reviewer #2: Yes

4. Have the authors made all data underlying the findings in their manuscript fully available?

Reviewer #2: Yes

5. Is the manuscript presented in an intelligible fashion and written in standard English?

Reviewer #2: Yes

6. Review Comments to the Author

Reviewer #2: (No Response)

7. PLOS authors have the option to publish the peer review history of their article (what does this mean?). If published, this will include your full peer review and any attached files.

Reviewer #2: No

---

## [Editor Report · Acceptance letter]

31 Jul 2024

PONE-D-24-15982R2 

PLOS ONE

Dear Dr. Ali, 

I'm pleased to inform you that your manuscript has been deemed suitable for publication in PLOS ONE. Congratulations! Your manuscript is now being handed over to our production team.

Kind regards, 

on behalf of

Dr. Tebelay Dilnessa 

Academic Editor

PLOS ONE